# Reassessing the Impact of Fear Appeals in Sustainable Consumption Communication: An Investigation into Message Types and Message Foci

Chih-Ching Yu * and Ching Lu

Department of Business Administration, National Taipei University, No. 151, University Rd., Sanxia Dist., New Taipei City 237303, Taiwan; jeanluscu@gmail.com
* Correspondence: yucc@mail.ntpu.edu.tw

**Abstract:** In the face of escalating environmental challenges, effective communication has assumed unprecedented significance. This study addresses a critical research gap concerning the utilization of fear appeal in promoting sustainable consumer behavior. Rather than treating fear appeals as a singular construct, this research decomposes them into distinct components to explore the nuanced impacts of various fear appeal aspects. Additionally, we investigate how different message foci influence the effectiveness of various fear appeals in promoting sustainable consumption. To achieve these objectives, we designed an experimental study encompassing eight distinct scenarios, achieved through the manipulation of four types of fear appeal and two types of message focus. Participants were randomly assigned to one of these scenarios. Our findings reveal noteworthy differences in purchasing intentions, particularly in the context of various fear appeal types. Specifically, we observed significant variations between the economic fear appeal and the physical fear appeal, as well as between the self-esteem fear appeal and the physical fear appeal. Surprisingly, our analysis of the moderating effect of different message foci did not align with our initial expectations. The outcomes of this research offer valuable insights for green marketers, empowering them to employ diverse facets of fear appeal in their marketing strategies with greater flexibility and efficiency. These findings underscore the evolving landscape of sustainable consumer behavior and the evolving role of communication in addressing environmental challenges.

**Keywords:** sustainable consumption; fear appeal; effective communication; sustainable consumer behavior; purchase intention

## 1. Introduction

The rapid progress of science and technology has effectively satisfied the increasing desires and needs of human beings, but often at the expense of the environment's carrying capacity [1]. With the popularization of the concept of global sustainable development, attention to sustainable consumption has increased. Effective communication now plays an essential role in mitigating the increasingly complex environmental challenges our world faces today. The United Nations' Sustainable Development Agenda 2030 Goal 12, calling for responsible consumption and production to improve the quality of life [2], coupled with the appearance of global environmental crises, has made communication in promoting sustainable consumption an emerging field of research [3]. However, consumers often lack sufficient information on sustainable consumption; for instance, to identify whether a product is associated with sustainable technology or values from product descriptions. As a result, there is a significant gap between consumers' environmental awareness and their sustainable consumption behaviors (e.g., [4–6], etc.). Developing effective communication strategies to enhance consumers' sustainable consumption has, therefore, become a critical issue for both academic and practical purposes.

Certain research has raised ethical concerns about the use of fear appeal as a communication strategy to promote recommended preventive behaviors, citing reasons such as its potential ineffectiveness, exacerbation of the complexity of the situation, limitations on freedom from fear, and the risk of adopting a paternalistic attitude leading to unintended negative consequences like denial, backlash, avoidance, defensiveness, stigmatization, depression, anxiety, increased risk behavior, and a sense of loss of control (e.g., [7–10]).

However, research also identifies fear as a potential driver of behavioral change by increasing people's perception of the severity and susceptibility of physical issues through heightened risk assessment, combined with boosting their self-efficacy and response efficacy regarding behavioral solutions (e.g., [10–12]). Consequently, the use of fear appeal as an effective communication tool has gained widespread recognition, particularly in areas such as healthcare, information security, politics, and environmental conservation (e.g., [13–17]).

Green advertisements that emphasize the sustainable value of products play a crucial role in product marketing [18]. When advertisements adopt fear appeal, they promote consumers' fear of the future and guilt for destroying the environment to influence consumers' behavioral intentions [19]. Because "messages intended to stir emotions are a central component of modern communication" [20], fear can increase campaign effectiveness in different contexts [21]. Nai and Maier [20] revealed that political candidates gain a comparative advantage by applying fear appeal as a driver during election campaigns. Pittman et al. [22] discovered that green advertisements that primarily promote compassion, care, and vision are not convincing to non-green consumers. In contrast, using negative emotional appeal in messages implies that individuals' welfare and survival are threatened leading to increased behavioral intention. Liang et al. [23] argued that advertisements eliciting negative emotions do not affect environmental protection behaviors, but increase green product purchase intentions. Talukdar and Lindsey [24] also reported that when fear is perceived, the asymmetric pattern of demand response sensitivity to price changes is weakened among consumers with regard to healthy versus unhealthy food. However, existing studies have largely regarded fear appeal as an overall construct. (e.g., [25–30]) and have overlooked its different aspects. Drawing from examples like Passyn [25] and Addo et al. [30], the first study explored a novel approach to motivating health protection when task difficulty is prominent by incorporating regret into fear appeals. The second study investigated the shifts in purchasing behavior in the aftermath of the COVID-19 outbreak in Wuhan, China, and globally, using the fear appeal theory as a framework. Both of these studies share the common approach of treating fear appeal as a holistic construct, without breaking it down into more specific components. This approach has limited the study of the effect of fear appeal; most studies have failed to address the diverse effects of the different aspects of fear appeal, which prevents businesses from flexibly utilizing different types of fear appeal in their marketing strategies. The specific research question of this study is how different types of fear appeal messages and message foci affect the purchase intention of green consumption. Moreover, studies on fear appeal have often employed students as samples (e.g., [28,29,31–33]), and their research findings may thus not be generalizable. The limited generalizability arises from the fact that students often belong to a narrow demographic, typically characterized by youth and higher education. Consequently, findings from such samples may not be readily applicable to the broader population, given that the characteristics and behaviors of students may not align with those of the entire population.

In response to the research question, this study applied four types of fear appeal (physical, social, economic, and self-esteem) observed in green advertising [34] to examine fear-appeal-based communication. Furthermore, we employed universal sampling to comprehensively analyze the differences in individuals' perceptions and responses according to different demographic factors when they encounter different sustainable topics. Moreover, studies on communication have often regarded regulatory focus as a crucial consideration [35–38]; however, studies on fear appeal have yet to integrate this concept. It is noteworthy that Regulatory Focus Theory is a framework that delves into how individuals

approach goal pursuit, encompassing their motivations and perceptions in the processes of judgment and decision making. It posits two separate and independent self-regulatory orientations: prevention and promotion [39]. Therefore, this study employed promotion and prevention focus, two types of message foci, in the manipulation design [40] to analyze fear appeal in different experimental settings. In addition, to prevent the preferences of specific groups from influencing the analysis of the experimental topic [41], this study used the food industry, the industry most consumers commonly engage with, for the empirical analysis. Our study of sustainable consumption communication aims to make the following theoretical and empirical contributions. First, to fill the research gap on sustainable consumption by exploring the effect of fear appeal messages with different aspects on consumers' intentions to purchase sustainable products. Second, to investigate how different message foci affect the impacts of different types of fear appeal on sustainable food consumption. Finally, we plan to provide a reference for enterprises that wish to apply fear appeals to practical strategies for promoting sustainable consumption.

## 2. Literature Review and Hypotheses

### 2.1. Communication Effectiveness

"Companies often struggle to find effective communication strategies that induce consumers to buy green products or engage in other environmentally friendly behaviors" [42]. The stimulus–organism–response (SOR) model [43] describes the responses triggered in humans through stimulation, with specific stimuli generating specific behavioral modes. This theory has also been extended to the field of marketing research. Kotler et al. [44] considered that the main factors affecting consumers' purchase decisions are their backgrounds (e.g., cultural, societal, individual, and psychological aspects) and decision-making processes. However, because neither of these factors is an observable behavior and the mind is a "black box", further analysis is required.

Regarding the theory of communicative action, Habermas and McCarthy [45] propose three principles for effective communication. First, both parties must hold equal positions. Second, the content of the communicated messages must be based on facts to encourage message receivers to accept the message content. Finally, message senders must express their beliefs, intentions, feelings, and desires to enhance the credibility of the message among message receivers. This theory emphasizes the equal positions of communicating parties and the achievement of mutual understanding.

### 2.2. Fear Appeal

In the 1980s, the idea of sustainable consumption as a part of the general consumption process gained considerable attention. The number of green consumers with environmental concerns has increased and various emotional appeals for environmental preservation have been used in commercial advertising [46]. Compared to other types of emotional appeals, fear-based appeals are more persuasive [47]. Pezzulo [48] discovered that perceived fear generated by individuals is related to individual differences in personal perception, imagination, and inference. Abdel-Khalek [49] determined that people in different communities have different factors that influence their fears. Although many studies on fear appeal have been conducted since the 1950s [10], most have regarded fear appeal as an overall construct rather than a multicomponent construct. This approach has hindered in-depth investigation of the actual effects of fear appeal.

Bartikowski et al. [34] divided fear appeals into being physical-, social-, economic-, and self-esteem-based. In this study, physical fear appeals were regarded as statements threatening individuals' bodies, health, or lives, whereas self-esteem-based fear appeals addressed the fear of tarnishing one's self-image in social contexts, leading to a loss of pride, self-respect, and psychological well-being [50,51]. DeWall and Pond Jr. [52] explained that social fear appeals are based on the fear of breaking the rules, breaking the law, or being isolated by groups and lacking a sense of belonging. LaTour and Zahra [53] described economic fear appeals as relating to losing income and the source of economic

income, which can generate anxiety. Tannenbaum et al. [54] conducted a meta-analysis to evaluate the influence of fear appeals on attitudes, intentions, and behaviors. Their results highlighted a positive impact of fear appeals on these factors. Moderation analyses, guided by established fear appeal theories, showed that the effectiveness of fear appeals increased when messages conveyed higher fear levels, included efficacy statements, and emphasized high susceptibility and severity. Empirically, Li et al. [55] introduced a social fear appeal research model by conducting an experiment to investigate how recipients' reactance proneness influenced their assessment of the threat and efficacy associated with addressing the issue of microplastic pollution in Taiwan. The study's findings indicate that fear-induced communication emerged as an effective persuasive approach, with the perceived threat playing a crucial role in the ability of fear appeal messages to achieve persuasive outcomes. In a separate study, Liu et al. [56] conducted research on physical fear appeals by investigating how the inclusion of social norm appeals (individual vs. group cues) and fear appeals in COVID-19 vaccine campaign posters affected perceived communication quality and vaccination intention. Their findings show that the presence of fear appeals in COVID-19 vaccine campaign posters resulted in reduced perceived communication quality and vaccination intention levels compared to posters that did not include fear appeals. Moreover, Sobol and Giroux [57] provided valuable insights into fear appeals, introducing a promising message-framing approach and uncovering a distinct mechanism. Their findings confirm the effectiveness of fear appeals in influencing consumer behavior, especially when emphasizing a nonspecific threat. While fear appeals have been preliminarily classified, comprehensive comparisons among the various types of fear appeals are still deficient.

### 2.3. Protection Motivation Theory, Conservation of Resources Theory, and Fear-Appeal-Based Communication

Regarding consumers' response mechanisms triggered by the negative pressure of fear appeals, Protection Motivation Theory (PMT) and Conservation of Resources Theory (COR) explain consumers' relevant behaviors (e.g., [58–62], etc.). PMT maintains that when individuals are afraid, their personal motivation to solve problems is triggered to prevent them from feeling threatened and fearful. This corresponds to self-protection behavior [63,64]. COR states that individuals assess their response behaviors according to their available resources. When individuals experience stress, they are motivated to obtain and preserve resources in order to prevent resource loss. Their stored resources can be divided into objects, states, individuals, and energy [65]. Fear messages stimulate consumers to protect, preserve, and guard their resources.

Shin et al. [28] adopted the extended parallel process model [66] to assess whether fear appeal advertisements generate negative advertisement attitudes and positive product attitudes, leading to increased product purchase intentions. However, the fear appeal messages used in the experimental design did not stimulate consumers' purchase intentions. In the same study, the researchers manipulated the sources of information and divided advertisement sponsors into non-profit and for-profit organizations. The results indicated that consumers' purchase intentions did not differ significantly. Lee et al. [32] performed experiments within global and local frameworks (i.e., global environmental issues vs. local environmental issues), and employed fear appeal messages and hope appeal messages to create four combinations of experimental scenarios. This study revealed that within the global framework, fear appeal leads to superior results. In the experiment with an environmental topic and a local framework, neither fear nor behavioral intention was generated by the participants. Shehryar and Hunt [67] applied terror management theory to demonstrate how the nature of threatening consequences in fear appeal messages affects responses to such communications. They differentiate between death-related and non-death-related consequences, offering insights into maladaptive responses to fear appeals. Beitelspacher et al. [68] investigate the financial costs of implementing radio frequency identification (RFID) technology by studying its impact on retailer–consumer relationships.

Using an experimental design, they found that trust in the retailer and privacy expectations influence threat perceptions among respondents, subsequently impacting attitudes and behaviors. Furthermore, Morales et al. [69] investigated the distinctive impact of disgust in persuasion. In a series of four studies, their findings revealed that incorporating disgust into a fear appeal significantly boosts message persuasion and compliance, surpassing appeals that evoke fear alone. This persuasive effect of disgust is attributed to its potent and immediate avoidance response.

*2.4. Using Fear Appeal for Sustainable Consumption*

In the context of using fear appeal to promote sustainable consumption, Chen [29] assessed the impact of climate change fear appeals on individuals' pro-environmental intentions and investigated factors influencing these intentions across different levels of fear appeal. The findings suggest that the readers of low-fear appeal text experienced greater fear and displayed a higher intent to engage in pro-environmental behavior compared to those exposed to high-fear appeal text.

Hunter and Röös [70] delved into the role of fear, particularly the danger control process, in the context of climate change and food choices, aiming to better comprehend the factors motivating consumers to reduce meat consumption. Their findings underscore the significance of enhancing consumers' self-efficacy in adopting meat alternatives and educating them about their role in mitigating the threat.

In addition, Shen and Kim [71], excluding the work of Shin et al. [28] mentioned earlier, explored the interplay between fear appeal intensity (moderate vs. high) and temporal frames (proximal vs. distal) in eco-friendly clothing advertising in China. The results indicated that consumers strongly connected to eco-friendly clothing held a more positive attitude towards advertising employing moderate-level fear appeals than high-level ones.

Our study referenced Bartikowski et al. [34] and employed the physical, social, economic, and self-esteem-based fear appeal types commonly used in fear appeal communication. For the study design, we integrated the mental black box of the SOR model as well as the idea that communicating with different people requires the use of different contexts, as described in the theory of communicative action. We proposed the following hypothesis:

**Hypothesis 1 (H1).** *The effect of fear appeal messages on consumers' intention to purchase sustainable products is influenced by the message type.*

*2.5. Effects of Fear Appeal Message Types*

Trope et al. [72] adopted a construal level theory and the perspective of psychological distance, using different psychological levels to explain individuals' performances and predict their behaviors. It is noteworthy that psychological distance refers to the perceived or subjective separation between an individual and an event, object, or concept. It is a psychological construct that encompasses different dimensions, including temporal distance, spatial distance, social distance, and hypotheticality [72]. Message events were divided into high-level and low-level events. High-level events involved relatively abstract constructs and depictions and relatively greater psychological distances, whereas low-level events involved highly detailed and clear descriptions and relatively shorter psychological distances. Simultaneously, time, space, social distance, and probability were used as structures to explain the relationships between such events. As this theoretical framework supports the explanation of psychological levels and behaviors, it is often used for the research manipulation of different scenarios and descriptions [73] and is valuable for the construction of messages [74]. Among the four fear appeal scenarios, each employing physical, social, economic, or self-esteem-based fear appeals, the physical scenario focused on individuals' physical conditions. The self-esteem-based scenario emphasized both personal perceptions and individuals' perceptions of themselves as viewed by other people. The economic scenario integrated overall market trends, and the social scenario focused on the survival of other people and species. Regarding spatial distance within

the construal level theory, compared with social, economic, and self-esteem-based scenarios, the physical scenario was the only scenario directly affecting the individual, and this scenario was thus a low-level event with a shorter psychological distance. Regarding temporal distance, in contrast to physical fear appeal, the other three fear appeal types required external conditions and longer periods of time, and were thus high-level events with further psychological distances. Message receivers can produce stronger perceptions and responses by higher-level content because such content has less noise interference [72]. Therefore, this study proposed the following three extended hypotheses:

**Hypothesis 1a (H1a).** *The effect of fear appeal on consumers' intention to purchase sustainable products is stronger in the social scenario than in the physical scenario.*

**Hypothesis 1b (H1b).** *The effect of fear appeal on consumers' intention to purchase sustainable products is stronger in the economic scenario than in the physical scenario.*

**Hypothesis 1c (H1c).** *The effect of fear appeal on consumers' intention to purchase sustainable products is stronger in the self-esteem-based scenario than in the physical scenario.*

*2.6. The Regulatory Effect of Message Foci*

In PMT, another factor that affects perceived fear is an individual's personality. Thus, this study referenced a regulatory focus theory (RFT) and designated two regulatory foci, namely prevention focus and promotion focus, to evaluate the effects of different scenarios. RFT claims that strong regulatory focus significantly affects the emotional responses generated in the process of achieving goals [40]. Higgins [39] stated that with promotion focus, people focus on the positive results they obtain when they succeed and ignore the lack of positive results they obtain when they fail. In contrast, with prevention focus, people focus on the riddance of negative results when they succeed and on the avoidance of negative results when they fail. In other words, people with promotion focus pursue the possibility of positive results, whereas those with prevention focus pursue the avoidance of negative results [39]. Empirically, Liberman et al. [75] discovered that people encountering different message foci respond differently. Zou and Chan [76] further identified that in green marketing, promotion and prevention focus generate different behavioral intentions, which was further explored in the present study in relation to fear appeal on sustainable consumption communication.

According to a regulatory fit theory based on RFT [77], people with different regulatory foci have stronger responses to messages with the same focus. Specifically, people with promotion focus have stronger responses to messages with promotion focus, and people with prevention focus have stronger responses to messages with prevention focus. Hence, this study proposes the following hypothesis:

**Hypothesis 2 (H2).** *Message types of fear appeal with different message foci have different effects on consumers' intention to purchase sustainable products.*

Hernandez et al. [78] integrated message types, message foci, and psychological distance into an analysis that revealed that high-level content and promotion focus have a favorable fit. By contrast, with low-level content, using a prevention focus can increase fit favorability and persuasion effects. According to H1, in which the social, economic, and self-esteem-based scenarios have a greater psychological distance than the physical scenario, if we integrate RFT and regulatory fit theory, we can infer that with the promotion focus, the social, economic, and self-esteem-based scenarios are more persuasive than the physical scenario. Based on the aforementioned theories, this study proposes the following three extended hypotheses:

**Hypothesis 2a (H2a).** *Social fear appeal has a stronger effect on consumers' intention to purchase sustainable products than physical fear appeal in the context of promotion focus compared with the effect observed in the context of prevention focus.*

**Hypothesis 2b (H2b).** *Economic fear appeal has a stronger effect on consumers' intention to purchase sustainable products than physical fear appeal in the context of promotion focus compared with the effect observed in the context of prevention focus.*

**Hypothesis 2c (H2c).** *Self-esteem-based fear appeal has a stronger effect on consumers' intention to purchase sustainable products than physical fear appeal in the context of promotion focus compared with the effect observed in the context of prevention focus.*

Based on the hypotheses, we established the following research framework (Figure 1).

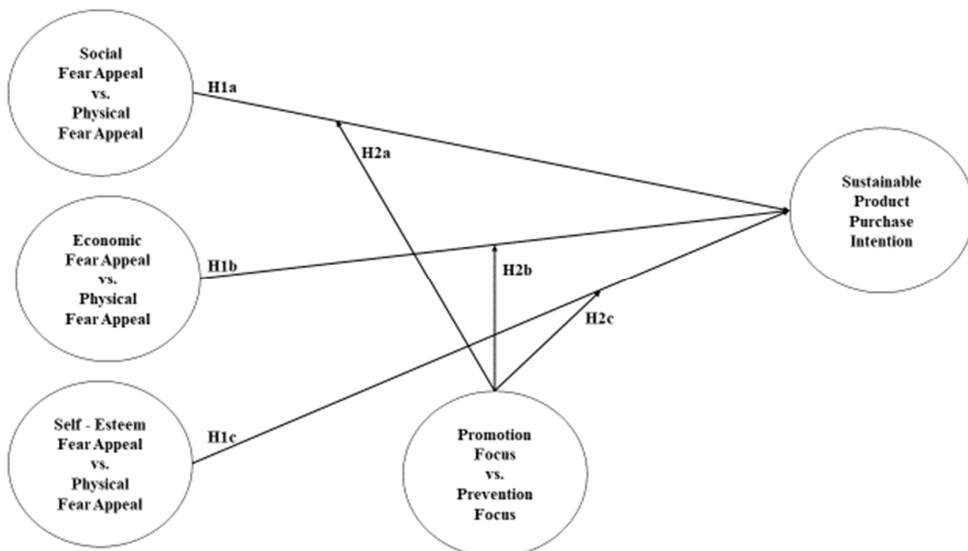

**Figure 1.** Research framework.

### 3. Methods

*3.1. Manipulations*

This study adopted the classification method of Bartikowski et al. [34] and integrated the physical-, social-, economic-, and self-esteem-based fear appeal types and promotion and prevention message foci to form eight scenarios. Items on purchase intention and demographic information were incorporated into these eight scenarios to generate eight versions of the questionnaires, which were labeled A to H. The questionnaire is included as Appendix A within the study. Table 1 presents the descriptions and examples of the eight scenarios.

**Table 1.** Experimental scenarios of the study.

| Experimental scenario #1 (Version A) Physical fear × promotion focus Example: If you do not adopt green food, you will be less healthy. | Experimental scenario #2 (Version B) Physical fear × prevention focus Example: If you do not adopt green food, you cannot remain healthy. |
|---|---|
| Experimental scenario #3 (Version C) Social fear × promotion focus Example: If you do not adopt green food, you cannot increase the overall social benefit. | Experimental scenario #4 (Version D) Social fear ×prevention focus Example: If you do not adopt green food, you cannot maintain the overall social benefit. |

**Table 1.** *Cont.*

| Experimental scenario #5 (Version E) Economic fear × promotion focus Example: If you do not adopt green food, your economic situation will deteriorate. | Experimental scenario #6 (Version F) Economic fear × prevention focus Example: If you do not adopt green food, you cannot maintain your economic situation. |
|---|---|
| Experimental scenario #7 (Version G) Self-esteem-based fear × promotion focus Example: If you do not adopt green food, you will lose your self-image. | Experimental scenario #8 (Version H) Self-esteem-based fear × prevention focus Example: Failing to select green foods could impact your ability to uphold your personal image. |

### 3.2. Scenario Content

During the experimental scenario design stage, we conducted three tests of fear perception to prevent the fear appeal of certain scenarios from being too low to achieve the experimental results. The first test was performed to determine whether participants felt fear in the scenarios. The second test, using the same fear appeal message types and two different message foci, explored whether participants had significantly different perceptions of fear. The third test explored whether participants perceived significantly different levels of fear when exposed to the four fear appeal types with one message focus. After multiple tests and feedback, we verified that the participants could no longer identify the differences between levels of fear immediately after reading the message text.

For the experimental content, we selected topics from daily life news events and magazine articles. Specifically, we selected food topics that had been verified and were relatively uncontroversial. The sources were *National Geographic Magazine* and a report from "Environmental Working Group 2019" [79]. In addition, to avoid the interference of elements, such as shapes and sounds, the contents of the scenarios in this study were presented using text. We presented the potential reasons for and hazards of not purchasing green food to ensure that participants had a sufficient understanding of the topic and could effectively interpret the content of the messages [80]. To ensure that the reading load in each scenario was consistent, the words for each scenario required approximately one minute of reading time. To ensure that the research content was relevant to real-world consumption situations, we integrated the Corpus of Contemporary American English and searched for contemporary wording related to physical, social, economic, and self-esteem-based aspects and to food topics when designing the experimental scenarios.

### 3.3. Measurement Variables

The questionnaire used in this study was divided into three parts. The first part was the reading of the design scenario, in which each participant was randomly presented with one of the eight scenarios. The second part related to the purchase intention. The last part gathered the demographic information of the participants, including sex, age, and educational attainment.

The purpose of measuring sustainable product purchase intention was to evaluate the probability of consumers purchasing sustainable products. The dependent variable's (i.e., "intention to purchase green food") score was the average of an individual's answers to all four questions presented in Part II of the questionnaire. We used the food industry as an empirical research subject. For the questionnaire items, we referenced and modified items from Cronin Jr. et al. [81], Dodds et al. [82], Venkatesh et al. [83], and Chan [84]. The questionnaire was scored on a 5-point Likert scale, which included the following options: "Strongly Disagree", "Disagree", "Neutral", "Agree", and "Strongly Agree".

### 3.4. Research Participants and Sampling

Fear appeal studies have frequently used narrow or inappropriate samples such as university students; consequently, conclusions drawn from studies with relatively homogeneous samples may not apply to other population groups [41]. Therefore, in this study, we surveyed people from different age groups. However, because minors have not yet

fully developed their personalities and comprehensive logical reasoning abilities, and because fear-based messages may cause negative emotional reactions, such as panic or hopelessness [85], we only surveyed participants aged 18 years or older.

The questionnaire was distributed using the online questionnaire design platform Survey Cake. In addition, the link was posted on Facebook fan pages. Additional experimental samples were obtained using snowball sampling. A questionnaire with one of the eight scenarios was randomly generated when a person clicked on the link. The questionnaires were distributed between December 2020 and February 2021. In total, 453 valid responses were obtained. Among the respondents, 49% were male and 51% were female. Most participants were aged between 18 and 25 years, and between 36 and 45 years. People of other ages accounted for at least 8% of the total sample. Over half of the participants had educational attainment of university or above. The statistics of sample structure are shown in Table 2.

**Table 2.** Statistics of the sample structure.

| Sample Attribute | | Number | Percentage |
|---|---|---|---|
| Sex | Male | 220 | 49% |
| | Female | 233 | 51% |
| Age | 18–25 years | 131 | 29% |
| | 26–35 years | 78 | 17% |
| | 36–45 years | 121 | 27% |
| | 46–55 years | 78 | 17% |
| | 56 years or older | 45 | 10% |
| Educational level | High school (or below) | 95 | 21% |
| | University | 234 | 52% |
| | Graduate school (or above) | 124 | 27% |

## 4. Analysis and Results

### 4.1. Testing the Different Effects of Message Types

A one-way analysis of variance (ANOVA) was applied to analyze purchase intentions under different scenarios (H1). However, this study did not meet the normality assumption, as indicated by a *p*-value of 0.000 (<0.05) for the Kolmogorov–Smirnov statistic. On the other hand, the study passed the homogeneity of variances test, with a *p*-value of 0.598 (>0.05) for the Levene statistic. It is worth noting that the ANOVA test is considered robust and can maintain the validity of probability statements even when assumptions like Normality and Equal Variances are violated [86]. The results indicated that the fear appeal message type significantly influenced consumers' sustainable product purchase intentions [$F_{(3, 449)} = 3.691$, $p = 0.012$]. This study employed a t-test to analyze H1a, H1b, and H1c in relation to the various scenarios and examine whether significant differences in purchase intentions exist. According to the statistical results presented in Table 3, the comparisons of the physical and social scenarios, the physical and economic scenarios, and the physical and self-esteem-based scenarios revealed that none of their upper and lower confidence intervals contained zero, and their p-values were all smaller than 0.01, indicating a significant moderating effect. Thus, H1a, H1b, and H1c were all supported.

**Table 3.** Results of the *t*-test of consumers' purchase intentions in the physical scenarios compared with that in the other scenarios.

| Message Type | LLCI | ULCI | *t*-Value | *p* |
|---|---|---|---|---|
| Physical, Social | −0.499 | −0.079 | −2.712 | 0.007 |
| Physical, Economic | −0.579 | −0.186 | −3.843 | 0.000 |
| Physical, Self-esteem-based | −0.558 | −0.161 | −3.561 | 0.000 |

LLCI = lower limit confident level; ULCI = upper limit confident level.

*4.2. Testing the Moderating Effect of Message Foci*

To verify whether message foci play a moderating role in the effect of fear appeal on consumers' intention to purchase sustainable products, this study used the physical scenario as the basis of comparison and converted the social, economic, and self-esteem-based scenarios into three groups of dummy variables (T1, T2, and T3, respectively). In other words, T1, T2, and T3 were jointly, rather than individually, adopted to denote four types of fear. The values used to represent physical, social, economic, and self-esteem fears on the three dummies, respectively, are specified as (0, 0, 0), (1, 0, 0), (0, 1, 0), and (0, 0, 1).

The moderated test here was adopted to assess whether or not message foci affect the effect of the differences between physical fears and the other fear types on purchase intention. The variables in the model used to assess the moderating effects include T1, T2, T3, message foci, T1 × message foci, T2 × message foci, T3 × message foci, and controls. The coefficient estimates of the three interaction terms represent the moderating effects. We used the Macro Process to estimate the model. The results suggest that the upper-limit confidence interval (UPCI) and lower-limit confidence intervals (LLCI) for the coefficient of T1 × message foci included zero (ULCI = 0.007, LLCI = −0.487), indicating a non-significant moderating effect. Furthermore, the UPCIs and LLCIs for the coefficients of T2 × message foci and T3 × message foci also included zero (ULCI = 0.275, LLCI = −0.202; ULCI = 0.444, LLCI = −0.039); therefore, neither had a significant moderating effect. Based on these statistical results, H2a, H2b, and H2c were all rejected. That is, compared with the physical scenario, in the social, economic, and self-esteem-based scenarios, the impact of fear appeal on consumers' sustainable product purchase intentions was not affected by the moderating effect of message foci.

## 5. Discussion

Our analysis indicates that the type of fear appeal message indeed exerts an influence on consumers' intentions to purchase sustainable products. Specifically, our results demonstrated noteworthy disparities in purchase intentions across different scenarios, thus providing robust support for the hypothesis regarding message type differences. Significant variations were observed in purchase intention within the social and physical scenarios (H1a), economic and physical scenarios (H1b), and self-esteem-based and physical scenarios (H1c), thereby confirming the validity of H1. This study not only reaffirms the impact of fear appeal messages on consumers' intentions to purchase sustainable products but also rectifies a limitation observed in prior research. Unlike previous studies that treated fear appeal as a singular construct, our research embraced the notion of fear appeal as a multi-component construct. By employing paired comparison analysis, we elucidated the distinct effects of various fear appeal message types on consumers' intentions to purchase sustainable products. Our findings revealed that, in comparison to physical fear appeal messages, those centered on social, economic, and self-esteem considerations demonstrated superior efficacy. Nonetheless, no significant differences were observed among these three types of fear appeal messages.

Regarding the analysis of the moderating effect of message foci (H2), we found that, when compared to the physical fear appeal message type, the effects of the social (H2a), economic (H2b), and self-esteem-based (H2c) fear appeal types on consumers' intentions to purchase sustainable products were not subject to moderation by different message foci. In simpler terms, varying message foci, whether promotional or preventive, did not yield distinct moderating effects on the influence of fear appeal on consumers' sustainable product purchase intentions. The unexpected nature of these results may be attributed to the boundary conditions for the regulatory fit effect. It implies that the effect of fit on judgment reflects a misattribution effect emerging from people's confusion about the source of their feeling to the characteristics of the target they are evaluating [87]. In essence, when people become aware of their emotional reactions and believe that these emotions could potentially bias their judgments, they may consciously adjust their judgments to counteract any potential bias [87]. This observation provides an intriguing angle for future research

on the interplay between message foci and fear appeal within the context of sustainable product purchase intentions.

On the one hand, while some research evidence suggests gender differences in the perception of fear appeals and their social consequences (e.g., [88–90]), our study did not yield similar results. On the other hand, we did identify significant age-related variations in the perception of fear appeals, impacting purchasing intention, specifically between the age groups of 18 to 25 and 36 to 45 ($p < 0.01$), as well as between the age groups of 18 to 25 and 46 to 55 ($p < 0.01$). An explanation for the age differences in the perception of fear appeals we observed could be that consumers in the two older age groups typically serve as the primary decision makers for household food shopping.

### 5.1. Implications

The results of this study have several theoretical and practical implications. Firstly, it contributes significantly to the understanding of fear appeal by dissecting its multifaceted nature. By examining the varying effects of different fear appeal message types on consumers' purchase intentions, our research provides valuable insights for managers seeking to employ fear appeals strategically. This knowledge aids in tailoring communication strategies, assessing product features, and optimizing the allocation of organizational resources to achieve the most effective communication outcomes. Secondly, this study addresses the potential biases that have affected previous fear appeal research, such as studies characterized by homogenous sample groups, vague target audiences, and external factors that may have influenced results. Furthermore, this study represents the first attempt in fear appeal research to integrate psychological distance, message type, and message foci. While the outcomes regarding the moderating effect of message foci may not have aligned with our initial expectations, they expand the horizons of fear appeal research, opening up possibilities for future investigations in this domain.

Regarding the texts used in the experimental design, we integrated the Corpus of Contemporary American English to employ scenarios that resemble real-world scenarios. Contemporary languages related to the research topic were identified using this corpus. We then used the food industry as an empirical research industry because consumers commonly interact with that industry. As studies on fear have often failed to quantify the intensity of fear in their experimental designs, in the present study, before the formal scenarios were formulated, we tested the perception of fear and modified the intensity of fear potentially generated in different scenarios to resolve the problem.

This research revealed that using certain fear appeal message types as stimuli (i.e., social, economic, and self-esteem-based types) was more effective than others (i.e., physical type). Therefore, we suggest that when sustainable marketers plan to employ fear-based marketing to communicate with consumers and boost their intent to purchase sustainable products, they should convey their message persuasively and, perhaps, in a commercial style. For example, when employing a social message type, they might say, "Make a choice that safeguards our collective well-being. Neglecting sustainable products puts our social benefits at risk. Your decisions impact our future—act now for a secure and sustainable society". Additionally, when sustainable marketers intend to utilize economic-based communications to engage with consumers, they can promote the message, "Don't let the cost of neglect haunt your legacy. Neglecting sustainable products can trigger resource price surges and severely impact vital industries like agriculture, forestry, fisheries, and husbandry. Choose wisely. Choose sustainability". Furthermore, when employing self-esteem-based communication, sustainable marketers can utilize the message, "Your self-image is at stake. Choosing non-green products may cast a shadow on our future. Choose sustainability for a brighter tomorrow!". Essentially, by employing these strategic communication approaches with consumers, businesses could effectively reinforce their commitment to purchasing sustainable products.

In conclusion, this study bridges a critical research gap within the realm of sustainable consumption. It achieves this by deconstructing fear appeals into distinct components,

rather than treating them as a singular construct. Additionally, it examines the multifaceted impact of various fear appeal elements on the purchasing intentions of sustainable consumers. This comprehensive understanding empowers sustainable marketers to employ fear appeal in a more flexible and efficient manner, adapting their strategies to different facets of this persuasive technique.

*5.2. Limitations and Future Research*

Initially, this study employed a text-based questionnaire design to investigate the influence of fear appeal on sustainable product purchase intention. Future research endeavors could consider incorporating multisensory experiences, including visual and auditory effects, to investigate the contemporary landscape of message transmission across diverse technological platforms [91]. Furthermore, when delving into the moderating variables associated with fear appeal and their influence on sustainable product purchase intentions, forthcoming studies may benefit from analyzing consumers' sustainable traits. These traits, which encompass dimensions such as altruism versus egoism and the extent of green preferences, can serve as effective segmentation variables. Such an approach could provide insights into the unexpected research findings concerning moderating variables uncovered in the current study. Moreover, to expand the scope of research in the field of fear appeals based on the foundations laid out in this study, future research could explore the potential influences of sub-cultures on fear appeals, such as distinctions between Eastern and Western cultural values.

We have identified several limitations in this study. Firstly, our sample population comprises individuals interested in green foods, but we lack precise information on the sample size due to a lack of practical data. Secondly, we used voluntary sampling, which may limit our sample, as participants in voluntary response samples tend to have strong opinions about the survey's subject matter. Finally, our study did not examine individual demographic variables in our model, such as male vs. female.

To date, theories on fear appeal cannot sufficiently explain the effectiveness of using emotional appeal to alter consumers' long-term behaviors [92]. Given the current lack of clarity surrounding the mechanisms behind various message types [93], there is a clear imperative for further research in this area. Moreover, in the context of promoting sustainable consumption, we propose the inclusion of older adults as research participants. With improvements in living conditions and advances in medical technology, older adults constitute an increasingly significant segment of the population in developed countries [94]. Considering their economic influence and health-related concerns, older adults hold substantial potential as a major green consumer group. Thus, investigating the use of fear appeal as a strategic communication tool for engaging with this demographic warrants comprehensive exploration.

**Author Contributions:** Conceptualization, C.-C.Y. and C.L.; Methodology, C.-C.Y. and C.L.; Project administration, C.-C.Y.; Investigation, C.-C.Y. and C.L.; Formal analysis, C.L.; Writing—original draft, C.L.; Writing—review and editing, C.-C.Y.; Validation, C.-C.Y.; Visualization, C.-C.Y. and C.L.; Supervision, C.-C.Y. All authors have read and agreed to the published version of the manuscript.

**Funding:** No external funding was received.

**Institutional Review Board Statement:** Not applicable.

**Informed Consent Statement:** Informed consent was obtained from all subjects involved in the study.

**Data Availability Statement:** The data presented in this study are available on request from the corresponding author. The data is not publicly available due to its use in ongoing research.

**Conflicts of Interest:** The authors declare no conflict of interest.

## Appendix A. Questionnaire

Dear Participants,

Hello! I am a graduate student from the Department of Business Administration at National Taipei University. Firstly, I sincerely appreciate your valuable time spent participating in this research survey.

This is an academic questionnaire designed to gain insights into consumer behaviors related to green food products. I would like to kindly ask you a few questions, and please be aware that there are no right or wrong answers; you only need to provide your personal opinions.

The data collected from this questionnaire will be used solely for academic research purposes. Your personal information will never be disclosed to third parties, so please feel confident in your responses. I am deeply grateful that you took the time to complete this questionnaire amidst your busy schedule. Your responses will greatly contribute to a better understanding of green food consumption, and I hope that this research will be beneficial to your future choices regarding green food products.

Best wishes for good health and well-being!

**May I ask if you are already 18 years old or not? ☐ Yes, I am. ☐No, I am not.**
**Part I. Scenario Statement.**

Please read the following statement.

(Note that when an individual clicks on the survey link, a questionnaire featuring one of the following eight scenarios will be randomly generated.)

**Experimental scenario #1 (Version A)**
**Physical fear × promotion focus**

Are you aware of what you consume daily?

Residues of heavy metals such as copper and zinc in seafood have the potential to cause lasting harm to your liver and kidneys. Consuming poultry or livestock of uncertain origin can facilitate the transmission of diseases. Even the fruits and vegetables that people typically regard as the healthiest may not always be free from risks. Strawberries are particularly prone to pesticide residues, and if not adequately washed, a single strawberry could contain a year's worth of pesticide.

Are you really aware of what you consume daily?

**Experimental scenario #2 (Version B)**
**Physical fear × prevention focus**

Have you eaten or have you not?

Are you truly eating the right way? In non-organic commercial farming practices that lack crop rotation and fallow periods, there is increased use of chemical fertilizers, leading to intensive cultivation.

The soil is deprived of organic elements, and the crops lack essential trace elements. A deficiency in trace elements can result in reduced metabolism, and a shortage of zinc and selenium can potentially hamper your immune system and antioxidant functions. With an insufficiency of trace elements and inadequate nutrition, 99% of modern individuals grapple with what is known as 'hidden hunger'.

**Experimental scenario #3 (Version C)**
**Social fear × promotion focus**

Have you ever considered that climate change could rob us of our most delightful culinary pleasures?

Avocado production is hindered by soaring temperatures, while bananas contend with relentless pest infestations in the blistering heat. Olive yields dwindle due to drought in their cultivation regions, leading to a substantial increase in the cost of everyday essentials.

When food transforms into a scarce and precious commodity, how many individuals will be compelled to struggle, even at the risk of their lives?

The trade in fish maw is an industry manipulated by South American criminal organizations, who employ it as a weapon alongside their drug trafficking operations. Will this remain an isolated occurrence, or is it a glimpse into the future?

**Experimental scenario #4 (Version D)**
**Social fear × prevention focus**

The bananas you enjoy are often the result of clearing land through deforestation, cutting down native tree species. Likewise, the coffee you savor might involve competing for cultivation space with gorillas. Cotton candy and chocolate, on the other hand, carry the aroma of rainforest slash-and-burn practices.

Agriculture and livestock stand as key contributors to the generation of greenhouse gases, which drive global warming. As ice sheets diminish and milder winters become more frequent, the ice-covered season for North America's five Great Lakes shortens by a day every two years. Just last year, Lake Erie's ice-covered season was reduced to a mere 18 days. Nevertheless, winter activities in northern latitudes could soon become the final remnants of our generation's history.

One can only ponder how many of the activities we currently take for granted will gradually find a place in the world's cultural heritage.

**Experimental scenario #5 (Version E)**
**Economic fear × promotion focus**

Overfishing in the oceans implies that, in just a few years, they will no longer yield substantial fish harvests. Uncontrolled land development and extensive farming practices have led to permanent soil and water contamination, resulting in chronic toxicity. The excessive deforestation of rainforests has reached a stage where they can no longer fully absorb carbon dioxide. To what extent is this orchestrated by profit-driven, unscrupulous corporate elites?

Do you still regard fair-trade products as mere marketing tactics to increase prices?

Do you believe that your supermarket choices have no impact on environmental sustainability? Should natural resources become depleted, can human economic and social activities persist?

**Experimental scenario #6 (Version F)**
**Economic fear × prevention focus**

The imported bananas you enjoy are contributing to poverty in their places of origin. The imported coffee you savor is incentivizing local farm owners at the source to employ more child labor, often in unlawful conditions.

Unfair trade practices are gradually undermining the economic structure of rural areas in the producing regions, exacerbating wealth disparities and deteriorating living conditions.

Once their very sustenance becomes a pressing issue at the source, the global economy becomes increasingly vulnerable, resulting in labor shortages, supply disruptions, fluctuations in futures markets, and inevitable price collapses.

**Experimental scenario #7 (Version G)**
**Self-esteem-based fear × promotion focus**

Are you aware of the inhumane conditions in which poultry and livestock farming occurs? Do you understand the harsh living conditions forced upon enslaved laborers in inhumane plantations? Are you acquainted with how rare ingredients are cultivated under the looming threat of criminal forces and their weapons?

Brutality, violence, and exploitation are merely the surface of the unfair trade product iceberg.

Can you honestly claim that your daily food consumption is untouched by these issues? Can you honestly assert that you are not one of the contributors to this blood-stained supply chain?

**Experimental scenario #8 (Version H)**
**Self-esteem-based fear × prevention focus**

Do you aspire to live a vibrant life, cherished by all? Are you still prioritizing visual appeal when selecting ingredients? Do you understand the distinction between organic products and in-conversion organic products? Can you differentiate between concentrated fruit juice and straight fruit juice? Do you still indulge in opulent, high-end cuisine?

If you lack even the most fundamental knowledge about green foods, I regret to say that you might be heading towards social isolation.

**Part II. Purchase intention for green foods.**

In this study, 'green foods' refers to food products that not only promote the preservation and enhancement of human health, societal, and ecological well-being but also possess specific certifications. These certified products include ISO-certified products, traceable agricultural products, Taiwan Quality foods, organic agricultural products, Fair Trade-certified products, and seafood certified by the Marine Stewardship Council.

|  | Strongly Disagree | Disagree | Neutral | Agree | Strongly Agree |
|---|---|---|---|---|---|
| It is very likely for me to buy green foods. | ☐ | ☐ | ☐ | ☐ | ☐ |
| I would recommend green foods to my friends. | ☐ | ☐ | ☐ | ☐ | ☐ |
| I would like to buy green foods. | ☐ | ☐ | ☐ | ☐ | ☐ |
| I plan to buy green foods in the near future. | ☐ | ☐ | ☐ | ☐ | ☐ |

**Part III. Individual information.**

| Gender | ☐Male | ☐Female | |
|---|---|---|---|
| Age | ☐18–25 | ☐26–35 | ☐36–45 |
| | ☐46~55 | ☐56 and above | |
| Occupation | ☐Business/Trade | ☐Financial/Insurance | ☐Technology R & D |
| | ☐Manufacturing | ☐Catering/Food | ☐Leisure/Entertainment |
| | ☐Clerical work | ☐Governemnt officer | ☐Media/Communication |
| | ☐Medical care | ☐Teacher | ☐House keeping |
| | ☐Retired | ☐Information Technology | ☐Student |
| | ☐Others | | |
| Education | ☐High school or below | ☐University | ☐Graduate school or above |
| Monthly Income | ☐TWD 25,000 or less ☐TWD 50,001~100,000 | ☐TWD 25,001~$50,000 ☐TWD 100,001 or more | |

Thanks so much for your participation!

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
