# Peer review of "Reassessing the Impact of Fear Appeals in Sustainable Consumption Communication: An Investigation into Message Types and Message Foci"

_sustainability, doi:10.3390/su152316442_

Round 1

Reviewer 1 Report

Comments and Suggestions for Authors

The scope of this piece seems more like an all-encompassing treatment of an entire dissertation. It would be better presented with a more focused view toward examining a single pairing of fear appeals, and this would allow the authors to better discuss each of the appeals with less haste. Because the article attempts to use four appeal types and two moderating factors, so much is presented that the reader loses focus easily. It would also help if the reader had an example of a fear scenario rather than a very brief (ten word or so) summary as is given in the piece. I know that the authors wish to protect content, but opening the content to the reader actually could result in more authors attempting to replicate the results, leading to better citation counts over time. As it is, we only glimpse what the authors are trying to say and prove without really understanding what participants were asked to evaluate.

Comments on the Quality of English Language

Pretty good. The rewrite of content will necessitate a further editing of language.

Reviewer 2 Report

Comments and Suggestions for Authors

This is an interesting research exercise but, in my view, additional issues need to be address:

  • Using fear appeals, whatever they nature – economic, social, etc.- entitles ethical issues that should be discussed as well. Reasons such as fear appeals being not effective, harming consumers, limiting freedom or being paternalistic, among others, stand to support ethicists’ concerns about fear appeals.  
  • The impact of the cultural context on the communication of fear appeals it is not discussed: experimental scenarios presented in Table 1 revolve around the individual, one’s own safety (except perhaps for scenarios 3 and 4). So, it is not very clear how the assumption that a person own’s interest go first would fit within a collectivist culture like the Chinese one, where  family/group interests are most likely to go before  the individual ones. Perhaps a clearer distinction between personal and cultural/social fear appeals scenarios should be made (both in the analysis and discussion of the results).
  •  
  • Research evidence points to gender differences in the perception of fear appeals and their social consequences; also, older adults seem to be more sensitive to positive messages and give preference to emotional goals. Data set includes these variables but there is no related discussion.
  • Discussion: any (Chinese) cultural values  emerging from the analysis?
  •  
  •  

Reviewer 3 Report

Comments and Suggestions for Authors

Comments on the Quality of English Language

Extensive editing of English language required. Especially, section 2.

Reviewer 4 Report

Comments and Suggestions for Authors

Dear authors,

Thank you very much for the opportunity to review your manuscript and share my comments to help improve the current version.

Overall, your research is interesting and represents a contribution to the field.

Some comments that you should please consider:

1)      Unfortunately, you uploaded the consent statement, but you did not share the questionnaire. The questionnaire represents one of your research's core elements as it mirrors your variables. Faulty questions would not, or only partially, represent the variable you want to survey.

2)      In your method part, you mention that you used as well snowball sampling. To how many people (with which demographics) did you send the questionnaire for snowballing? What were the criteria did you apply for these first samples?

3)      You received a valid sample of 453 responses. What is the overall number of people the sample represents? You need to prove the statistical relevance of your sample. Please add.

4)      As your sample method is voluntary sampling, your sample is very limited. That is ok, but you need to address that in the study limitations. Participants in a voluntary response sample usually choose to respond to surveys because they have a strong opinion on the subject of the survey. That is a significant limitation.

5)      Under 5.1 Implications, you recommend certain messages that for marketing purposes, should be used. These should be further discussed, and practical messages with examples should be provided. The message «If you do not purchase sustainable products, you will not be able to maintain your self-image» is not very practical and should be further discussed.

6)      Your references are mainly dated before 2020; none are from 2022 / 2023. Please expand your literature accordingly and consider the latest publications in the field.

Good luck and kind regards,

Round 2

Reviewer 1 Report

Comments and Suggestions for Authors

I did not appreciate the condescending tone of the reply to the author, but the changes suggested were made. Thank you for the inclusion of the questionnaire. It clarified greatly the direction of the study. While still sprawling in scope, this is a better paper than the first two versions.

Reviewer 2 Report

Comments and Suggestions for Authors

I am satisfied with the authors' responses to my comments.

Author Response

Dear Reviewer,

We express our heartfelt gratitude for your invaluable insights and valuable suggestions.

Best Regards,

Reviewer 4 Report

Comments and Suggestions for Authors

Dear authors,

Thank you for the implementation of my comments.

Kind regards
